# Kaposi's sarcoma-associated herpesvirus promotes mesenchymal-to-endothelial transition by resolving the bivalent chromatin of PROX1 gene

**Yao Ding**[1☯], **Weikang Chen**[1☯], **Zhengzhou Lu**[1], **Yan Wang**[1,2], **Yan Yuan**[1,3]*

**1** Institute of Human Virology, Zhongshan School of Medicine, Sun Yat-sen University, Guangzhou, Guangdong, China, **2** Guanghua School of Stomatology, Sun Yat-sen University, Guangzhou, Guangdong, China, **3** Department of Basic and Translational Sciences, University of Pennsylvania School of Dental Medicine, Philadelphia, Pennsylvania

☯ These authors contributed equally to this work.
* yuan2@upenn.edu

**Data Availability Statement:** All relevant data are within the manuscript and its Supporting Information files.

## Abstract

Increasing evidence suggests that Kaposi's sarcoma (KS) arises from Kaposi's sarcoma-associated herpesvirus (KSHV)-infected mesenchymal stem cells (MSCs) through mesenchymal-to-endothelial transition (MEndT). KSHV infection promotes MSC differentiation of endothelial lineage and acquisition of tumorigeneic phenotypes. To understand how KSHV induces MEndT and transforms MSCs to KS cells, we investigated the mechanism underlying KSHV-mediated MSC endothelial lineage differentiation. Like embryonic stem cells, MSC differentiation and fate determination are under epigenetic control. Prospero homeobox 1 (PROX1) is a master regulator that controls lymphatic vessel development and endothelial differentiation. We found that the PROX1 gene in MSCs harbors a distinctive bivalent epigenetic signature consisting of both active marker H3K4me3 and repressive marker H3K27me3, which poises expression of the genes, allowing timely activation upon differentiation signals or environmental stimuli. KSHV infection effectively resolves the bivalent chromatin by decreasing H3K27me3 and increasing H3K4me3 to activate the PROX1 gene. vIL-6 signaling leads to the recruitment of MLL2 and SET1 complexes to the PROX1 promoter to increase H3K4me3, and the vGPCR-VEGF-A axis is responsible for removing PRC2 from the promoter to reduce H3K27me3. Therefore, through a dual signaling process, KSHV activates PROX1 gene expression and initiates MEndT, which renders MSC tumorigenic features including angiogenesis, invasion and migration.

## Author summary

Numerous parallelisms between development and cancer led to the concept that cancer is a development problem over the past 50 years. As our knowledge of epigenetic regulation is advancing, the similarities between development and cancer are becoming more apparent, providing further support to the theory. KSHV infection of mesenchymal stem cells

**Funding:** This work is supported by grants from the Natural Science Foundation of China (81530069, 81772177) (YY). The funders had no role in study design, data collection and analysis, decision to publish, or preparation of the manuscript.

**Competing interests:** The authors have declared that no competing interests exist.

(MSCs) may result in Kaposi's sarcoma (KS) through mesenchymal-to-endothelial transition (MEndT), a process resembling endothelial differentiation during development. KSHV initiates MEndT by activating the homeobox gene PROX1, a master regulator of the lymphatic endothelial cell differentiation, at the epigenetic level. Here we found that the PROX1 gene resides in bivalent domain chromatin in MSCs and KSHV infection resolves it through a dual signaling process to activates the PROX1 gene, which initiates MEndT and confers MSC KS-like phenotypes. The significance of this study is two-fold. First, the study elucidated the mechanism underlying KSHV-mediated MEndT and KS development at the transcription level. Second, KSHV uses two independent pathways to elevate activating histone modification and decrease repressive marker, respectively, to resolved bivalent chromatin, revealing a two-factor-authentication mechanism in the epigenetic regulation, which may grant a more efficient and accurate response to activate a gene in bivalent chromatin.

## Introduction

Kaposi's sarcoma-associated herpesvirus (KSHV), also termed human herpesvirus type 8 (HHV8), has been proven to be the etiological agent of Kaposi's sarcoma (KS) [1], primary effusion lymphoma (PEL) [2], and multicentric Castleman's disease (MCD) [3]. Recently, KSHV was also found to be associated with childhood osteosarcoma in Xinjiang Uyghur population [4]. However, the lack of comprehensive understanding about KSHV infection and tumorigenesis hampers our ability to treat KSHV-associated cancers. For example, KS is atypical cancer. The onset of KS as simultaneously multiple lesions (in the absence of obvious metastasis), the multiclonal malignant nature (especially in the late stages), and the presence of unusual inflammatory cell infiltration and neoangiogenesis in the very early stage of KS, distinguish KS from other orthodox tumors. The origin of KS spindle cells remains elusive. The current leading model is that KS spindle cells may derive from endothelial cell lineage as KS cells express endothelial markers. However, KS cells are poorly differentiated and also express other markers such as smooth muscle, macrophage, and mesenchymal markers, suggesting that KS cells do not faithfully represent the endothelial cell lineage [5]. Recently, we found a series of evidence suggesting that KS originates from oral mesenchymal stem cells (MSCs) through KSHV-induced mesenchymal-to-endothelial transition (MEndT) [6].

MSCs have been identified as a population of hierarchical postnatal stem cells with the potential to self-renew and differentiate into osteoblasts, chondrocytes, adipocytes, cardiomyocytes, myoblasts, and neural cells [7,8]. The oral cavity contains a variety of distinct MSC populations, including dental pulp stem cells (DPSCs), periodontal ligament stem cells (PDLSCs), and gingiva/mucosa-derived mesenchymal stem cells (GMSCs) [9–11]. Among these MSCs, PDLSCs and GMSCs have the potential to directly interact with oral cavity saliva, microbiota, and virus, therefore having a great chance to be infected by KSHV in the oral cavity. Our recent study found that AIDS-KS spindle cells express Neuroectodermal stem cell marker (Nestin) and oral MSC marker CD29, suggesting an oral/craniofacial MSC lineage of AIDS-associated KS [6]. Furthermore, MSCs were found highly susceptible to KSHV infection, and the infection effectively promotes multiple lineage differentiation, especially endothelial differentiation. When implanted in mice, KSHV-infected MSCs were transformed into KS-like spindle-shaped cells with other KS-like phenotypes [6,12]. These findings provided evidence that KS derives from KSHV-infected MSCs. In addition, others identified PDGFRA+ MSCs as KS progenitors [13].

We sought to understand how KSHV infection regulates the MEndT process to promote malignant transformation. It is known that the regulation of stem cell fate determination and differentiation is mainly at the epigenetic level, which has been intensively studied in embryonic stem cells [14]. Embryonic stem cells undergo global remodeling during early stem cell development that commits them to the desired lineage. The lineage commitment of stem cells is controlled by epigenetic mechanisms, and histone modification appears to be the most important layer in such regulation [15]. Some critical transcription factor gene promoters in embryonic stem (ES) cells harbor a distinctive histone modification signature that combines the activating histone H3K4me3 mark and the repressive H3K27me3 mark. These bivalent domains are considered to poise the expression of developmental genes, allowing rapid activation upon developmental signals [16,17]. Increasing evidence suggests that mesenchymal stem cell differentiation, like embryonic development regulation, is also controlled at the epigenetic level [18]. Since MSCs face the same challenge as embryonic stem cells—multi-lineage differentiation and stem cell fate determination, it is possible that similar epigenetic mechanism found in embryonic stem cells may also apply to MSCs. The homeobox gene Prospero homeobox 1 (PROX1) is a master regulator gene that controls lymphatic vessel development and endothelial differentiation [19] and is up-regulated by KSHV infection in MSCs [6]. Elucidation of the regulation mechanism of PROX1 gene expression by KSHV is vital for comprehension of KSHV-mediated MEndT that leads to KS.

In the current study, we found that the PROX1 gene in MSCs exhibits a "bivalent" epigenetic signature consisting of both H3K4me3 and H3K27me3 marks. We investigated how the bivalent chromatin structure is established and how KSHV infection resolves the bivalent domain to activate PROX1 and MEndT genes. This study elucidates the mechanism underlying KSHV-mediated MEndT at the transcription level.

## Results

### Kaposi's sarcoma lesions express PROX1 and KSHV infection of mesenchymal stem cells induces PROX1 expression

It was reported that among 30 oral Kaposi sarcomas (KS) lesions, twenty-eight (93.3%) were positive for PROX1 [20]. To confirm that KSHV infection is responsible for the induction of PROX1 expression, we examined three cases of KS, one oral and two skin lesions for expression of PROX1 and KSHV antigen LANA using immunohistochemistry analysis. The result showed that all three samples were PROX1-positive and LANA positive (Fig 1A). Periodontal ligament stem cells (PDLSCs) were infected with KSHV and the expression of PROX1 in response to KSHV infection was examined by double-staining immunofluorescence assay (IFA) and Western analysis. IFA showed that PROX1 expression was consistently correlated with KSHV infection (LANA expression) (Fig 1B), and Western blot analysis showed that PROX1 expression was upregulated by KSHV (Fig 1C). Furthermore, mock- and KSHV-infected PDLSCs were implanted in immunocompromised mice under the kidney capsule. After four weeks, the implants were subjected to an immunohistochemistry study for PROX1 expression. As shown in Fig 1D, PROX1 expression was found to be significantly elevated in KSHV-infected PLDSCs in comparison to mock-infected PDLSCs. Overall, PROX1 expression is upregulated both *in vivo* and *in vitro* in response to KSHV infection in MSCs.

### PROX1 expression is essential for KSHV-induced mesenchymal-to-endothelial transition

PROX1 is the master regulator of endothelial differentiation and lymphatic vessel development. The expression of PROX1 in KS lesions and the induction of PROX1 expression by

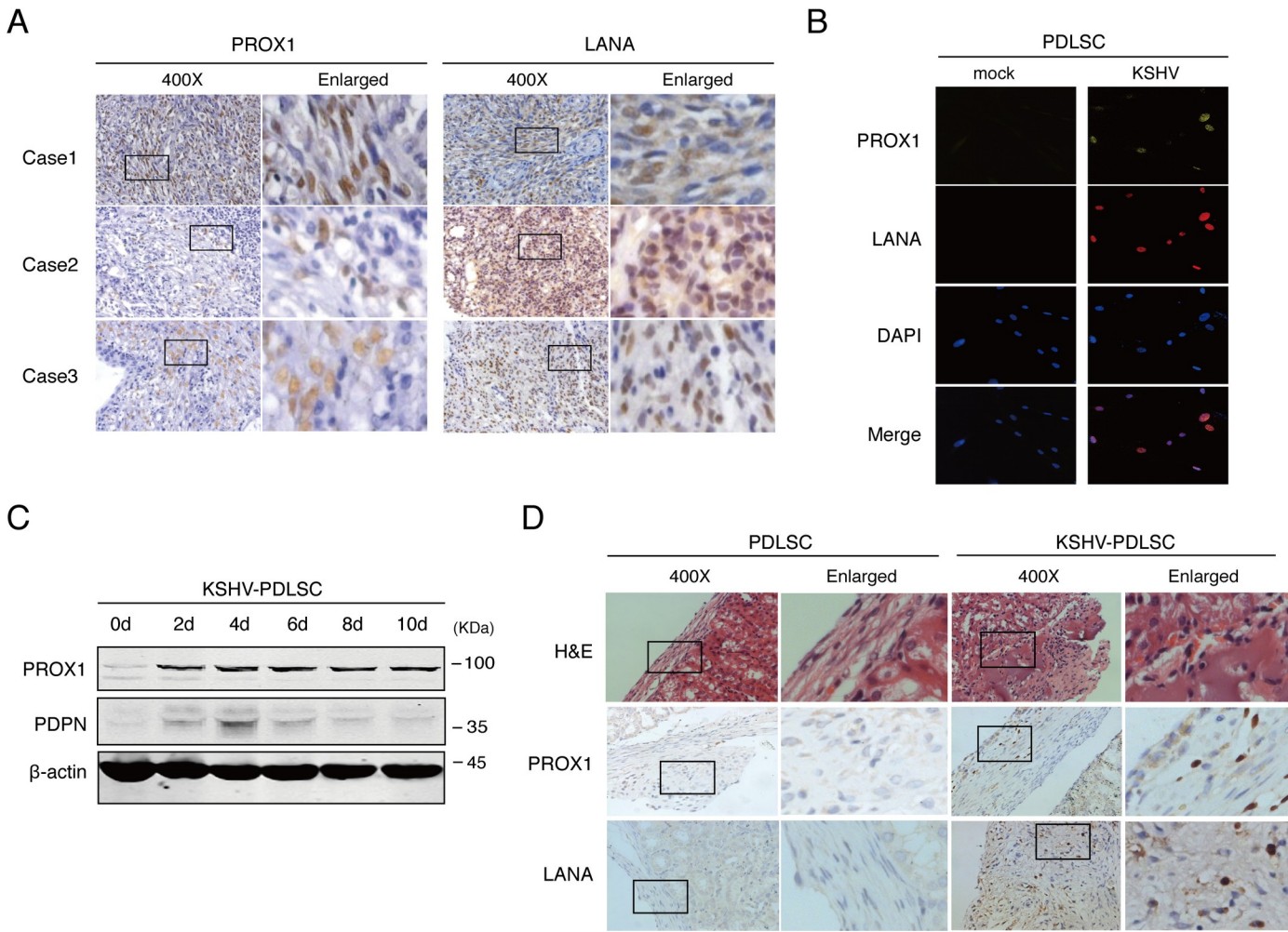

**Fig 1. Kaposi's sarcoma lesions express PROX1 and KSHV infection of mesenchymal stem cells induces PROX1 expression.** (A) Paraffin-embedded sections of Kaposi sarcoma (KS) lesions from three AIDS-KS patients were analyzed by immunohistochemistry (IHC) staining (400x and enlarged) for PROX1 and LANA. (B) Mock- and KSHV-infected PDLSCs were examined the expression of PROX1 and LANA by immunofluorescence assay (IFA). (C) Time courses of the expression of PROX1 and PDPN in PDLSCs after KSHV infection. (D) mock- and KSHV-infected PDLSCs were transplanted in mice under kidney capsules. After 4 weeks, PROX1 and LANA expression in the implants was analyzed by IHC.

KSHV infection in PDLSCs suggest that PROX1 plays a critical role in KSHV-mediated MEndT and KS development. To investigate this hypothesis, we determined the role of PROX1 in MEndT by using an shRNA-mediated gene silencing approach. An shRNA specific to PROX1 was introduced into PDLSCs by a lentiviral vector, and then the transduced PDLSCs were infected by KSHV. The knockdown efficiency of the shRNA was validated by Western blot (Fig 2A). The contribution of PROX1 to endothelial lineage differentiation and neovascularity was evaluated by Western blot to test endothelial markers expression and a Matrigel tubulogenesis assay. Results showed that the expression level of lymphatic endothelial markers (VEGFR3, LYVE-1, and PDPN) and pan-endothelial marker (VEGFR2) decreased in PDLSCs transduced with shPROX1 compared to those with control shRNA (Fig 2B). Correspondingly, shRNA-mediated PROX1 inhibition resulted in the loss of tubulogenesis (Fig 2C). Transwell-matrigel assay was used to assess cell invasion driven by PROX1. The result showed that shRNA-mediated PROX1 inhibition significantly reduced invasion ability compared to control shRNA (Fig 2D). Furthermore, when implanted in mice under kidney capsule, KSHV-

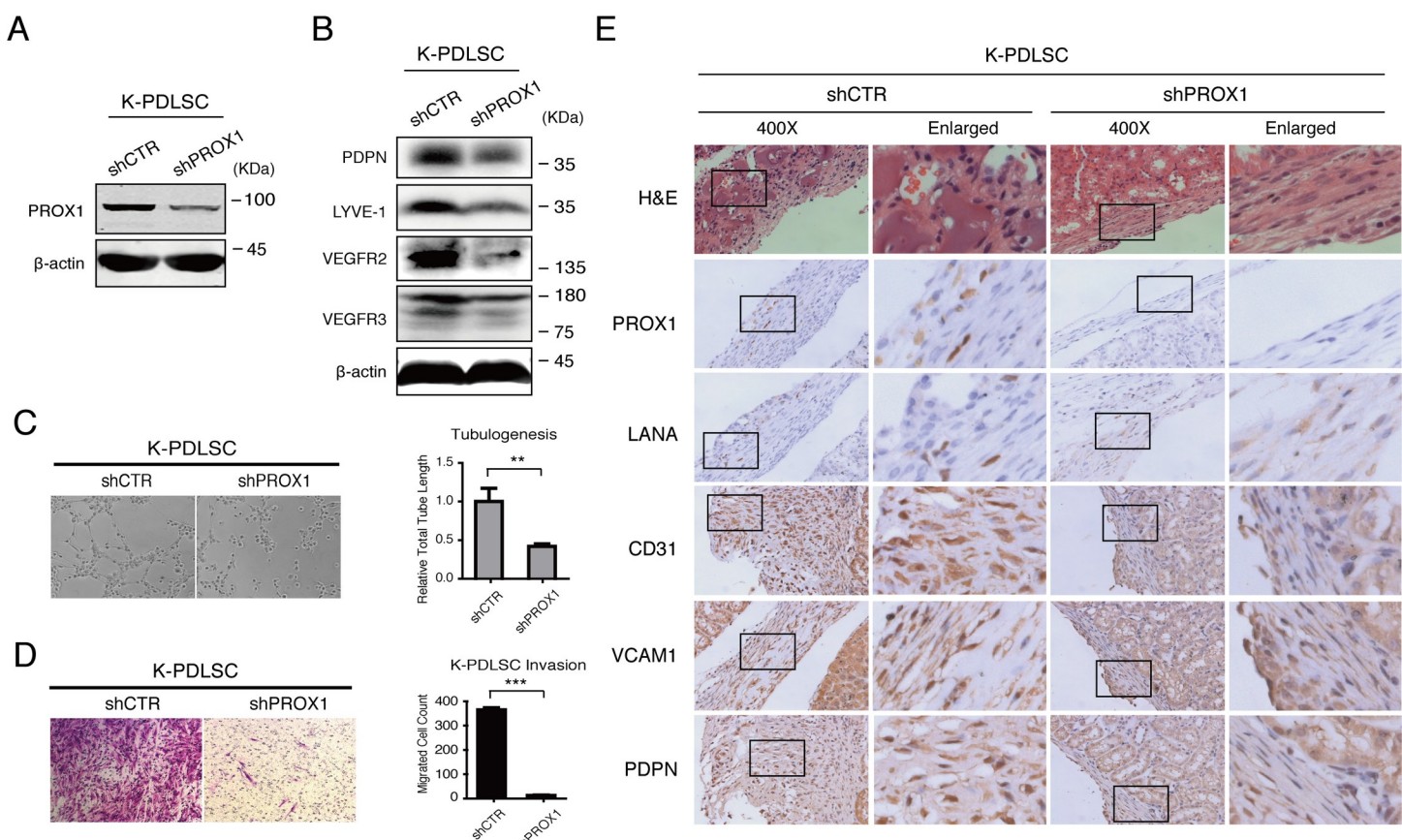

**Fig 2. The critical role of PROX1 in mesenchymal-endothelial transition induced by KSHV.** An shRNA lentivirus specific to PROX1 (shPROX1), along with a control shRNA (shCTR), were transduced into PDLSCs, followed by infection with KSHV. (A) The efficiency of PROX1 silencing was verified by Western blot. (B) The effects of the shRNA on the expression of pan-endothelial and lymphatic endothelial markers VEGFR2, LYVE-1, PDPN, and VEGFR3 were examined by Western blot. (C) KSHV-infected PDLSCs (K-PDLSC) transduced with shPROX1 or shCTR were evaluated for endothelial differentiation and angiogenic activities through a tubule formation assay. (D) K-PDLSCs transduced with shPROX1 or shCTR were subjected to cell invasion assay to determine the effect of PROX1 silencing on cell migration and invasion ability of K-PDLSCs. (E) KSHV-PDLSCs transduced with shPROX1 or shCTR were implanted under the kidney capsule of immunocompromised mice for 28 d. Then the kidney capsule was taken out and analyzed by hematoxylin and eosin (H&E) and IHC (PROX1, LANA, CD31, VCAM1, and PDPN) staining.

infected PDLSCs underwent an MEndT process revealed by the expression of endothelial markers CD31, VCAM1, and PDPN, while MEndT did not occur in PROX1-knockdown cells as they failed to express those endothelial markers (Fig 2E), suggesting that PROX1 is essential for the initiation of MEndT induced by KSHV infection.

## KSHV promotes MEndT through epigenetic regulation of the PROX1 gene that possesses a bivalent chromatin structure

We sought to understand how KSHV infection regulates the MEndT process. Evidence suggests that mesenchymal stem cell differentiation is controlled at the epigenetic level [18]. KSHV-mediated MSCs reprogramming likely takes place at this level. Histone modifications play critical roles in regulating the expression of developmental genes in mammalian development [21,22]. To explore epigenetic regulation of KSHV-mediated MEndT, we examined histone modifications on the chromatin of several MEndT genes in mock- and KSHV-infected PDLSCs using Chromatin immunoprecipitation (ChIP) assay. Results revealed an interesting feature that the PROX1 promoter locus in mock-infected PDLSCs heavily consists of both

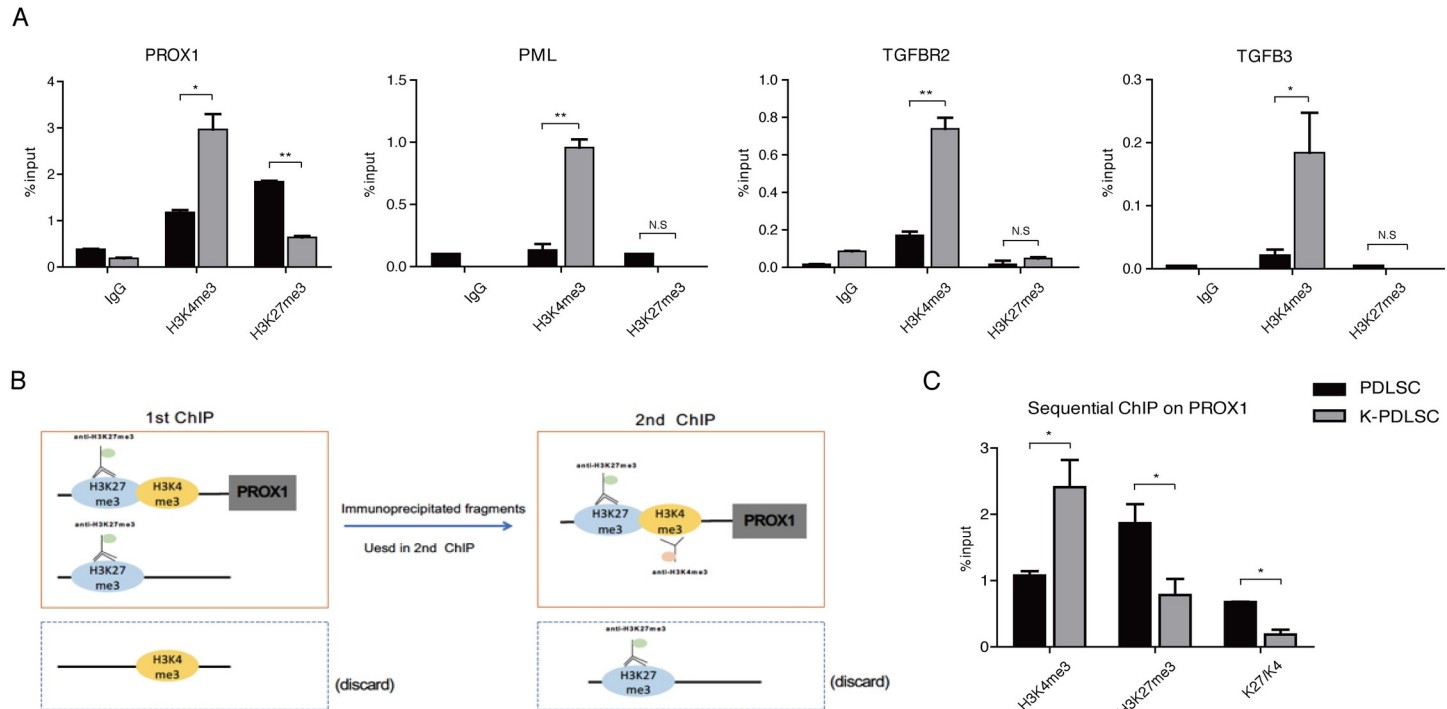

**Fig 3. PROX1 gene is harbored in a bivalent chromatin structure and KSHV infection can resolve it to activate PROX1 expression.** (A) Mock- and KSHV-infected PDLSCs were subjected to a Chromatin immunoprecipitation (ChIP) assay with antibodies against H3K4me3 and H3K27me3. The extracted DNA was analyzed by qPCR for the PROX1 promoter region and other MEndT-related genes, including PML, TGF-βR2 and TGF-β3. PROX1 gene in mock-infected PDLSCs heavily consists of both active mark H3K4me3 and repressive mark H3K27me3, exhibiting a distinctive bivalent chromatin domain. (B) Schematic illustration of the sequential ChIP assay. Cross-linked chromatin from MSCs was immunoprecipitated with an anti-H3K27me3 antibody followed by a second immunoprecipitation with an anti-H3K4me3 antibody. (C) The extracted DNA was analyzed by qPCR with primers of the PROX1 promoter, and the result showed that H3K4me3 and H3K27me3 co-exist in the same nucleosome in the PROX1 promoter. Error bars represent SD. Statistics analysis was performed using one-way ANOVA test, and P-value was calculated by GraphPad Prism. P-value < 0.05 was considered significant (*P<0.05, **P<0.01 and ***P<0.001) and N.S represented no significance.

repressive mark H3K27me3 and active mark H3K4me3. KSHV-infection dramatically increased H3K4me3 and decreased H3K27me3 in the PROX1 promoter (Fig 3A). This chromatin structure pattern, known as bivalent chromatin structure, has been reported in the genes of key transcriptional factors responsible for development in embryonic stem cells [16]. The bivalent domain structure was found to be restricted to the PROX1 promoter, not seen in the coding region of PROX1 (Supporting Information, S1 Fig). In contrast, other MEndT relevant genes, such as PML, TGF-β3, and TGF-βRII, obtained elevated H3K4me3 in the chromatin of their regulatory regions in response to KSHV infection (Fig 3A).

Given the unique nature of the bivalent domain on the PROX1 gene, we would confirm that the observed bivalent structure truly reflects the simultaneous presence of both H3K4me3 and H3K27me3 on the same chromosome rather than the presence of two subpopulations with different epigenetic characters. To this end, we performed a sequential ChIP assay in which PDLSCs chromatin was immunoprecipitated first with the Lys27 tri-methyl antibody and secondly with the Lys4 tri-methyl antibody (Fig 3B). This sequential purification is designed to retain only chromatin that concomitantly carries both the active and repressive histone modifications. Then real-time PCR was used to detect PROX1 promoter DNA on the bivalent nucleosomes. As shown in Fig 3C, H3K4me3 and H3K27me3 were found to co-exist on the same PROX1 promoter chromatin, accurately representing a true bivalent epigenetic state. Overall, our result suggests that the bivalent epigenetic feature may silence the PROX1 gene and its downstream cascade in MSCs but keep them poised for activation. KSHV

infection disrupts the balance and MSCs quickly turn on the PROX1 gene and downstream cascade, leading to MEndT.

## KSHV resolves the PROX1 bivalent chromatin by recruiting MLL2 and SET1 complexes to and removing PRC2 from the PROX1 promoter

To understand how KSHV manipulates PROX1 epigenetic regulation, we aimed to investigate the mechanism of establishing the bivalent domain in the PROX1 promoter and resolving it by KSHV. The central players in setting up and maintaining bivalent chromatin structure are the trithorax group (TrxG) and polycomb group (PcG) proteins [17]. The former consists of SET1A/B and MLL1-4 complexes, catalyzing the trimethylation of histone H3 Lys4 [23]. The latter forms multi-subunit polycomb-repressor complexes (PRC) 1 and 2, responsible for the trimethylation of histone H3 Lys27. PRC2 is particularly crucial for H3K27me3 in many developmental genes. To identify the tri-methyltransferase complexes that are recruited to PROX1 chromatin and altered in response to KSHV infection, ChIP assays were performed on mock- and KSHV-infected PDLSCs with antibodies against CFP1, MLL1, 2, respectively. The results showed that (i) prior to KSHV infection, MLL1 had been recruited to the PROX1 promoter region; (ii) KSHV infection did not change MLL1 content in the chromatin but recruited MLL2 and CFP1 to the promoter (Fig 4A–4C). CFP1, a CXXC finger protein 1 and specific component of the SET1A/B complex, serves as global regulation on H3K4 methylation. Thus, the elevation of CFP1 (SET1/COMPASS) suggests that KSHV infection increases the total level of H3K4me3 on the PROX1 promoter, and the recruitment of MLL2 (MLL2/COMPASS) resolves the bivalent domain of the PROX1 gene. Besides, no change in UTX (MLL3/4 COM-PASS) was detected in the PROX1 gene after KSHV infection (Fig 4D). RBBP5 and WDR5, shared by all six methyltransferase complexes (SET1A, SET1B, and MLL1-4), did not respond significantly to KSHV infection with a slight increase of RBBP5 (Fig 4E) and a minor decrease of WDR5 (Fig 4F). Therefore, we conclude that KSHV infection results in the recruitment of MLL2 and SET1 complexes to the PROX1 promoter region to elevate active histone marker H3K4me3.

On the other hand, the histone methyltransferase responsible for the deposition of the H3K27me3 mark on a bivalent domain is the PRC2 complex [24–26]. Our result showed that KSHV infection led to the removal of EZH2, the catalytic subunit of the PRC2 complex, from the PROX1 promoter, causing the loss of the H3K27me3 mark in the PROX1 region (Fig 4G). Altogether, KSHV-mediated recruitment of MLL2 and SET1 and removal of PRC2 resolve the bivalent domain of the PROX1 promoter.

## Identification of KSHV components that are responsible for epigenetic regulation of PROX1 gene expression

To elucidate the mechanism underlying KSHV altering the epigenetic features of the PROX1 gene, we sought to identify the viral genes that control PROX1 activation. Inspection of KSHV transcription profiles of KS lesions [27] and KSHV-infected PDLSCs [6] revealed a class of KSHV genes expressed in both KS lesions as well as KSHV-infected MSCs. We ectopically expressed these KSHV genes, including latent genes LANA, K12, and vFLIP and lytic genes K1, ORF45, ORF50, K8, PAN, vIL-6, and vGPCR in PDLSCs. Among these viral genes, vIL-6 and vGPCR significantly promoted PROX1 expression (Fig 5A).

The KSHV-encoded homolog to human interleukin-6 (vIL-6) could transmit its signals through gp130 to initiate proinflammatory and neoangigenesis and be sufficient to induce LEC-specific markers—PROX1 and PDPN [28–30]. Viral G-protein coupled receptor (vGPCR) could upregulate PROX1 during LEC reprogramming and render cells enhanced cell

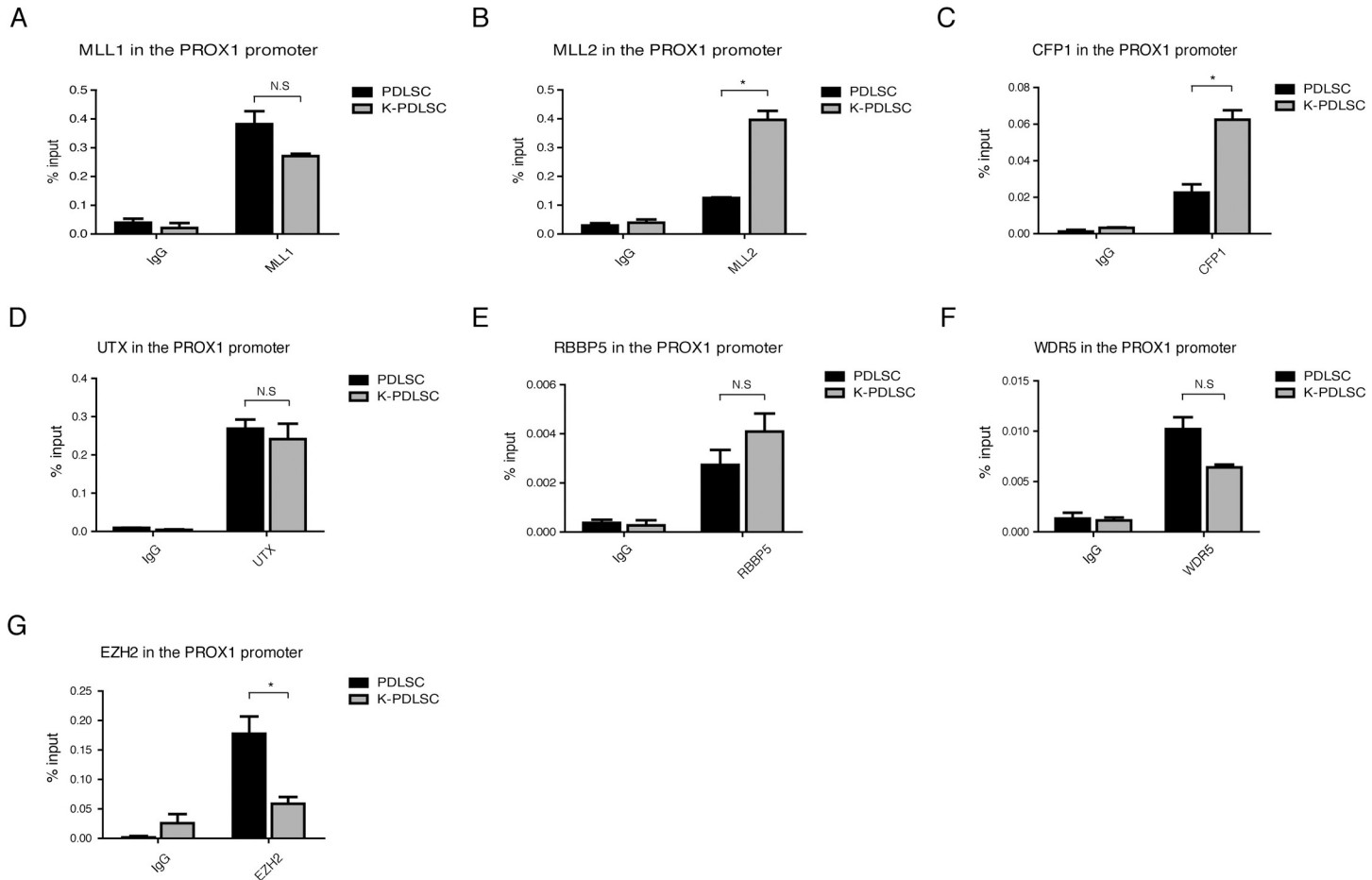

**Fig 4. KSHV resolves the PROX1 bivalent chromatin by recruiting MLL2 and CFP1 and removing PRC2 to the PROX1 Promoter.** Mock- and KSHV-infected PDLSCs were subjected to ChIP assay with antibodies against different histone methyltransferase components, including MLL1, MLL2, CFP1, UTX, RBBP5, WDR5, and EZH2 (a component of PRC2). Error bars represent SD. Statistics analysis was performed using the one-way ANOVA test. P-values < 0.05 were considered significant (*P<0.05, **P<0.01 and ***P<0.001) and N.S represents no significance.

proliferation, migration, and tumor formation [31]. To explore if vIL-6 and vGPCR promote PROX1 expression at the epigenetic level, we examined the effect of vIL-6 and vGPCR expression on histone modifications in the PROX1 gene in PDLSCs. We found that vIL-6 expression led to a significant increase in H3K4me3 marks but had no effect on H3K27me3 content, while vGPCR expression caused a dramatic decrease in H3K27me3 marks in the PROX1 promoter but did not change the level of H3K4me3 (Fig 5B and 5C). Co-expression of vIL-6 and vGPCR reproduced the alterations of H3K4me3 and H3K27me3, closely resembling to what was observed in KSHV-infected PDLSCs, and complete resolution of the bivalent domain (Fig 5D). Furthermore, vIL-6 expression deposited H3K4me3 via recruiting MLL2 histone methyltransferases to the promoter, and vGPCR removed H3K27me3 from the bivalent domain by downregulating PRC2 complex (EZH2) in the PROX1 promoter (Fig 5E and 5F). Taken together, KSHV resolves the PROX1 bivalent domain through two pathways, namely vIL-6 and vGPCR signaling, to alter active H3K4me3 and repressive H3K27me3, respectively, and consequently regulate PROX1 gene expression and MEndT process.

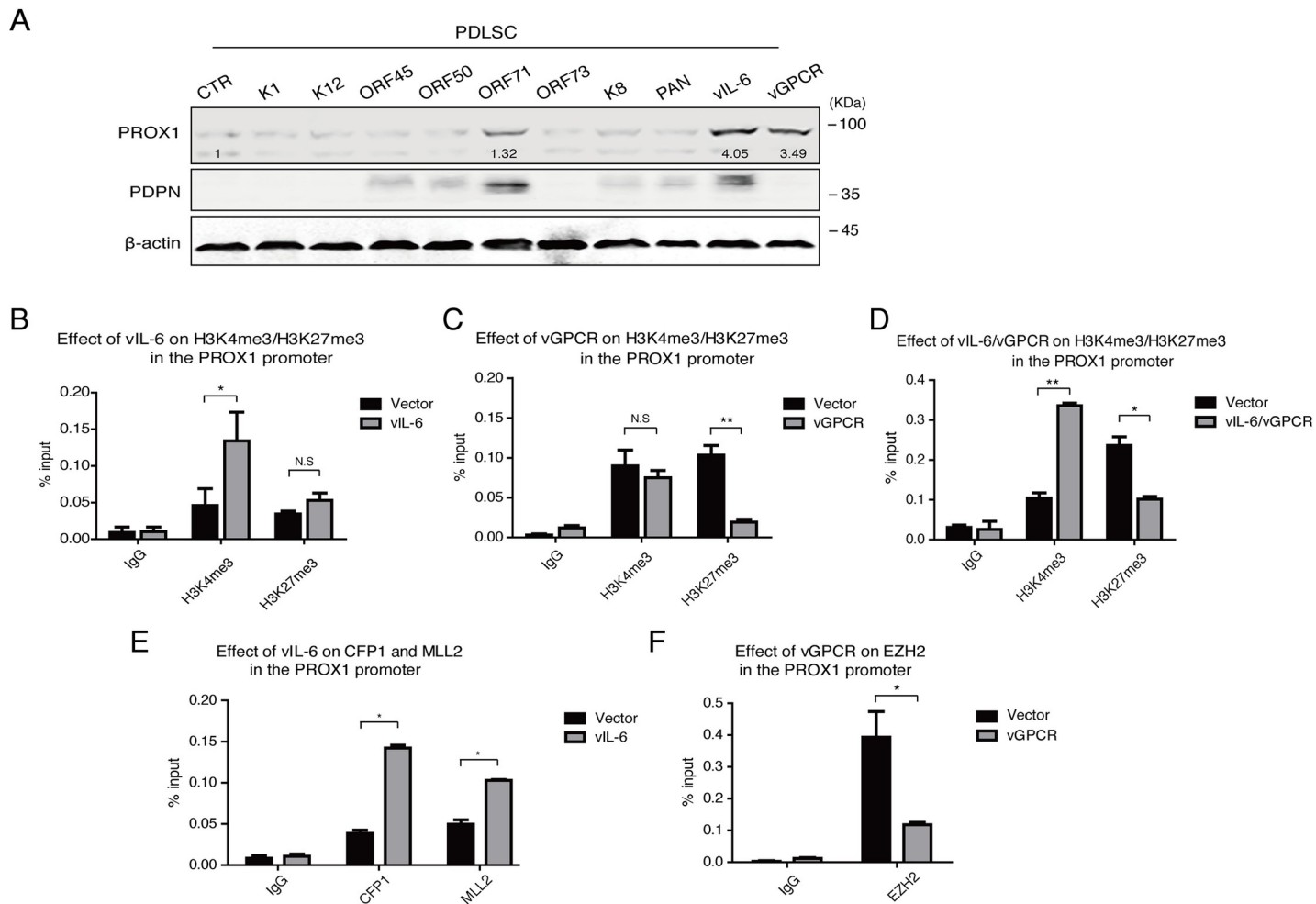

**Fig 5. Identification of KSHV components that are responsible for the epigenetic regulation of PROX1 gene expression.** (A) PDLSCs were transfected with a class of viral gene expression vectors, including K1, K12, ORF45, ORF50, ORF71, ORF73, K8, PAN, vIL-6, and vGPCR. The effects of these viral components on PROX1 expression were assayed by Western blot. ChIP assays were performed with PDLSCs ectopically expressing vIL-6 (B), vGPCR (C), or both (D) for their effects on H3K4 and H3K27 trimethylation in the PROX1 promoter. The effects of expression of vIL-6 (E) and vGPCR (F) on CFP1, MLL2, and EZH2 in the PROX1 gene were examined by ChIP assays. Error bars represent SD. Statistics analysis was performed using the one-way ANOVA test. P-value < 0.05 was considered significant (*P<0.05, **P<0.01 and ***P<0.001) and N.S represented no significance.

### KSHV-mediated vIL-6 and VEGF secretions regulate MSC differentiation by resolving the bivalent chromatin structure of the PROX1 gene

KS is an angiogenesis tumor, and pathological neoangiogenesis is a hallmark of the cancer. We previously reported that the treatment of MSCs with the conditioned medium of KSHV-infected MSCs sufficiently renders cells increased angiogenesis activity and MEndT, indicating that KSHV can induce MSC MEndT in a paracrine or autocrine manner [6,32]. We examined the conditioned medium of KSHV-infected PDLSCs for its effect on PROX1 expression and found that the treatment of PDLSCs with the conditioned medium dramatically upregulated PROX1 expression in PDLSCs, especially 48 hours post-infection (48 hpi) (Fig 6A). Furthermore, the treatment with the conditioned medium effectively altered the histone modification pattern closely resembling what was observed in KSHV-infected PDLSCs, i.e., increased H3K4me3 and decreased H3K27me3 in the PROX1 promoter region (Fig 6B).

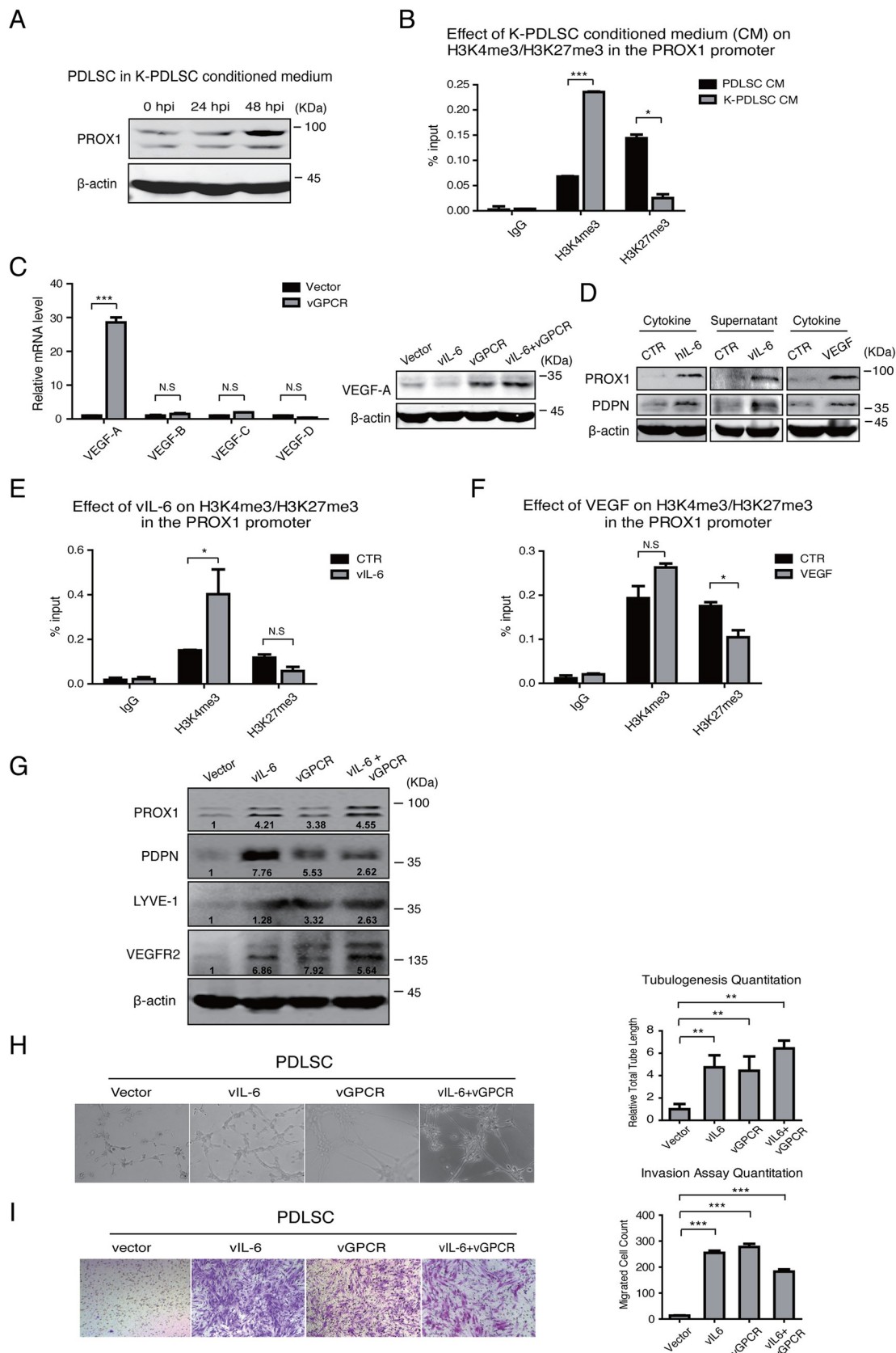

**Fig 6. Cytokines mediate the epigenetic regulation of PROX1 expression in KSHV-infected PDLSCs.** (A) PDLSCs treated with conditioned media of KSHV-infected PDLSC at indicated time points were examined for PROX1 expression by Western blot. (B) The conditioned medium-treated cells were subjected to a ChIP assay for histone modifications (H3K4me3 and H3K27me3) in the PROX1 promoter. (C) vGPCR induced the production of VEGF-A, revealed at mRNA and protein levels. (D) Human interleukin-6 (hIL-6), viral interleukin-6 (vIL-6) (supernatant of HEK293T cells transfected with a vIL-6 expression vector), and VEGF were added in PDLSC cultures, and cells were analyzed by Western blot with PROX1 and PDPN antibodies. ChIP assays were performed with vIL-6- (E) and VEGF-treated PDLSCs (F) for H3K4me3 and H3K27me3 modifications in the PROX1 promoter. (G) The expression of endothelial differentiation markers PDPN, LYVE-1, and VEGFR2 in vIL-6, vGPCR, or vIL-6/vGPCR expressing PDLSCs was evaluated by Western blot. (H) vIL-6, vGPCR, and vIL-6/vGPCR expressing PDLSCs were placed on Matrigel, and tubulogenesis was examined by Tubule formation assay and quantified by measuring the total segment tube length using the ImageJ software. (I) vIL-6, vGPCR, and vIL-6/vGPCR expressing PDLSCs ($1.5 \times 10^4$ cells/well) were seeded in the upper chamber of Transwell with Matrigel in serum-free α-MEM. α-MEM containing 20% FBS was used as a stimulus for chemotaxis in the lower chamber. Cells migrated to the lower chamber were stained with crystal violet and counted. Quantification was performed by the ImageJ software. The error bars represent SD. Statistics analysis was performed using t-test in GraphPad Prism. P-values $< 0.05$ were considered significant (*$P<0.05$, **$P<0.01$ and ***$P<0.001$) and N.S represented no significance.

In light of that KSHV-encoded vIL-6 and vGPCR can resolve the bivalent chromatin domain of the PROX1 gene, and so does the conditioned medium of KSHV-infected MSCs, we proposed a model for the paracrine regulation of epigenetic property in PROX1 gene as follows. It has been reported that vGPCR upregulates VEGF production to induce angiogenesis and lymphatic reprogramming [31,33]. Thus, vGPCR-induced VEGF, together with vIL-6, may be necessary and sufficient to resolve the bivalent chromatin and activate the PROX1 gene. We found that ectopic expression of vGPCR indeed induced VEGF-A expression in PDLSCs, as revealed by RT-qPCR and Western blot analysis (Fig 6C). When PDLSCs were treated by human interleukin-6 (hIL-6), vIL-6 supernatant, and VEGF-A, we observed the up-regulation of PROX1 expression and PDPN expression (Fig 6D). The effects of vIL-6 and VEGF-A on histone modification of the PROX1 gene were analyzed by treating PDLSCs with the supernatant of vIL-6-transfected 293T cells (48 hours post-transfection) and VEGF-A, followed by ChIP assay for H3K4me3 and H3K27me3. Interestingly, the treatment with vIL-6 supernatant led to increased H3K4me3, and VEGF-A caused decreased H3K27me3 in the PROX1 promoter (Fig 6E and 6F). Therefore, we conclude that KSHV induces PROX1 activation through a dual signaling process, i.e., activating vIL-6 signaling to increase active histone marker H3K4me3 and using vGPCR-VEGF-A axis to reduce repressive histone marker H3K27me3, to resolve the bivalent chromatin structure and activate PROX1 gene expression.

Then, the contribution of vIL-6 signaling and vGPCR-VEGF axis to the MEndT process and tumorigenic properties were evaluated. PDLSCs were transfected with vectors for vIL-6 and vGPCR and MEndT was monitored by the expression of lymphatic endothelial and pan-endothelial markers PDPN, LYVE-1, and VEGFR2 with Western analysis. Result showed that the endothelial markers were significantly elevated in response to vIL-6 and vGPCR (Fig 6G). In addition, in vIL-6 or/and vGPCR-overexpressed PDLSCs, both angiogenesis activity, represented by the formation of capillary-like tubules (Fig 6H), and malignant invasive feature, revealed by a Transwell-matrigel invasion assay (Fig 6I), were significantly elevated. Furthermore, a neutralizing antibody against VEGF-A can efficiently block endothelial marker expression and inhibit cell angiogenesis (tubule formation) and malignant invasion abilities (Supporting Information, S3 Fig). Taken together, vIL-6 and vGPCR-VEGF axis are crucial for MEndT and malignant transformation potential.

## vIL-6 signaling and vGPCR-VEGF axis are indispensable for epigenetic regulation of PROX1 and acquisition of tumorigenic properties

To assess the significance of vIL-6 and vGPCR-VEGF axis-mediated epigenetic regulation of PROX1, we silenced these two viral genes individually or simultaneously in KSHV-infected

PDLSCs by using small interference RNA (siRNA). Knockdown of vIL-6 and vGPCR, each moderately inhibited PROX1 expression compared to control siRNA (si-NC) cells (Fig 7A and 7C). The inhibition of vIL-6 reduced H3K4me3 marks in the PROX1 promoter but did not affect the H3K27me3 marker (Fig 7B), while knockdown of vGPCR expression increased H3K27me3 in the promoter but had no effect on H3K4me3 (Fig 7D). Double knockdown of vIL-6 and vGPCR dramatically reduced PROX1 expression (Fig 7E) by decreasing active mark H3K4me3 and elevating repressive mark H3K27me3 in the PROX1 promoter, counteracting the effect of KSHV infection in the PROX1 promoter (Fig 7F).

Then vGPCR-VEGF axis signaling was blocked in vGPCR and vIL6-expressing PDLSCs using an anti-VEGF-A neutralizing antibody and the effect of ablation of vGPCR axis on cell oncogenic phenotypes was examined. As demonstrated by tubule formation and Transwell-matrigel invasion assays, ectopical expression of vGPCR or co-expression of vGPCR and vIL-6 significantly elevated PDLSC angiogenesis and malignant invasion capabilities, which were completely abolished when the cells were incubated with anti-VEGF-A antibody (Fig 7G and 7H).

## Discussion

We previously reported that KSHV infection of human MSCs triggers an MEndT process that may lead to Kaposi's sarcoma development [6]. The current study sought to elucidate the mechanism of KSHV-mediated MEndT. The central event in KSHV-initiated MEndT is viral-mediated gene expression reprogramming that re-routes the differentiation process of MSCs. In this study, we found that the expression of the PROX1 gene, a master regulator of endothelial lineage differentiation and MEndT, is controlled at the epigenetic level through a bivalent chromatin domain that is usually seen in embryonic development transcription factor genes [16]. KSHV infection activates PROX1 by resolving the bivalent domain through a dual signaling process, i.e., vIL-6 signaling increases active histone marker H3K4me3, and vGPCR-VEGF axis decreases repressive histone marker H3K27me3. As a consequence, the activation of PROX1 initiates endothelial lineage differentiation and MEndT, leading to MSC acquisition of tumorigenic features such as proliferation, angiogenesis, and migration/invasion (schematically depicted in Fig 8).

Postnatal stem cells have multi-lineage differentiation potential into osteoblasts, adipocytes, chondrocytes, or myocytes [34]. PROX1 is a homeodomain transcription factor essential for the development of a variety of organs, including the lymphatic system, lens, heart, liver, pancreas, and central nervous system [35,36]. During early lymphatic development, endothelial cells in the cardinal vein exhibit a mixed phenotype of BECs and LECs. A subset of venous endothelial cells begins to express PROX1 and migrates out to form initial lymphatic vessels [19]. Overexpression of PROX1 in blood vascular endothelial cells (BEC) induces lymphatic vascular endothelial cells (LEC)-specific gene transcription in the BECs [37]. KSHV infection of BECs and LECs reprograms both types of cells to move away from their original cell identities and acquire the opposite cell type (LEC and BEC, respectively) properties, in accompanying with up- and down-regulation of PROX1 expression, respectively [38,39]. Further study delineated the opposing regulation of PROX1 by KSHV to direct cell differential fate through IL-3R (up-regulation) and Notch (down-regulation) signaling [40]. In our previous study, we found that KSHV-infection of oral MSCs can up-regulate PROX1 and, as a consequence, promote endothelial-lineage differentiation or MEndT, while KSHV-infection of terminally differentiated LECs down-regulates PROX1 [6]. Taken together, PROX1 is a key regulator in KSHV-mediated MEndT that reprograms MSCs leading to KS sarcomagenesis. In the current study, we confirmed the critical roles of PROX1 in developing KS tumorigenic phenotypes

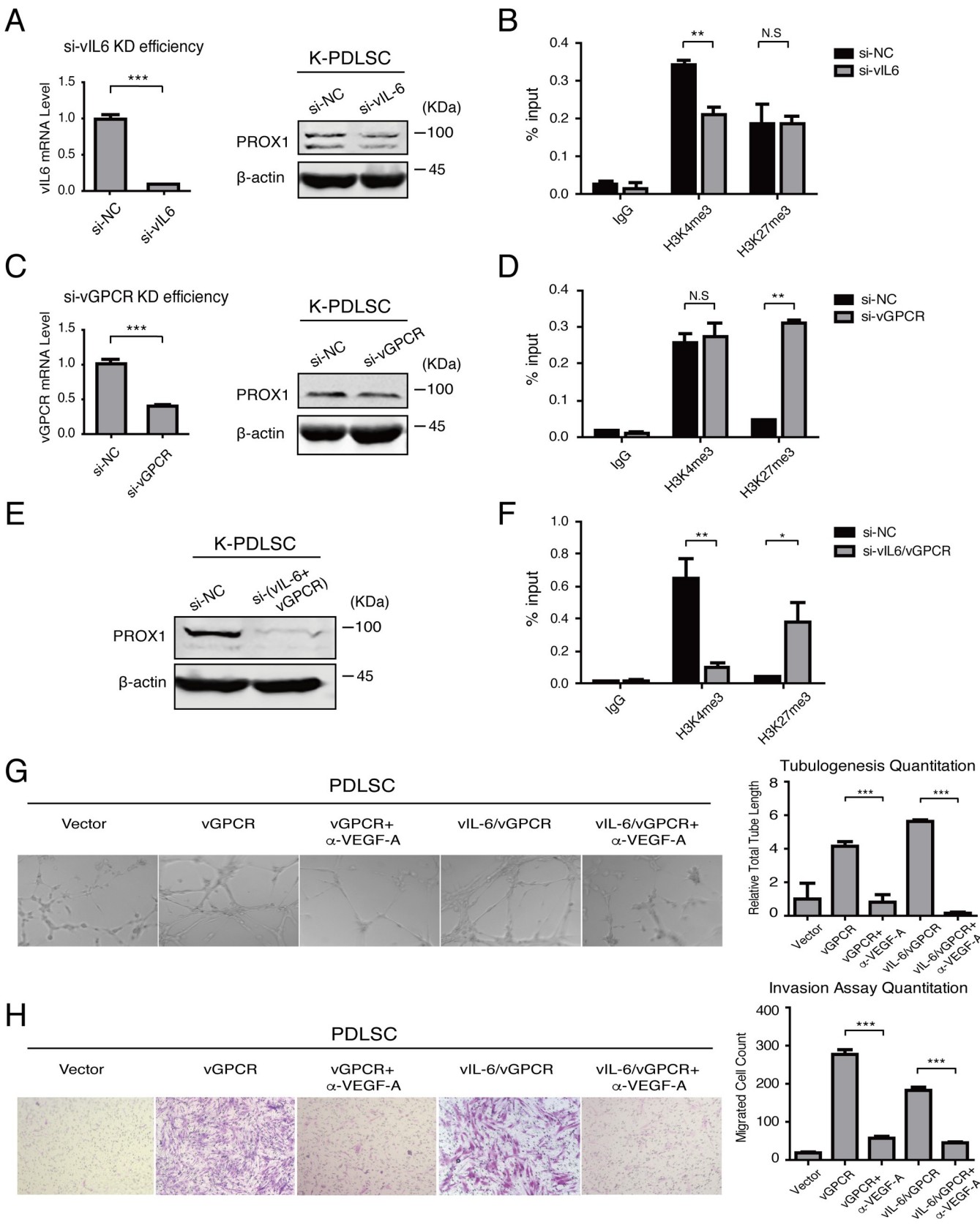

**Fig 7. vIL-6 signaling and vGPCR-VEGF axis contribute to KS tumorigenic properties.** (A) vIL-6 expression was silenced with siRNA (si-vIL-6 and negative control si-NC) in KSHV-infected PDLSCs. The knockdown (KD) efficiency was verified with quantitative RT-PCR and the effect on PROX1 expression was assayed by Western blot. (B) KSHV-PDLSCs transduced with si-vIL-6 or si-NC were subjected to ChIP assays to evaluate H3K4me3 and H3K27me3 in the PROX1 promoter. (C) vGPCR expression was silenced with a specific siRNA (si-vGPCR) in KSHV-infected PDLSCs. The knockdown efficiency was verified with quantitative RT-PCR and the effect on PROX1 expression was assayed by Western blot. (D) KSHV-PDLSCs transduced with si-vGPCR or si-NC were subjected to ChIP to evaluate H3K4me3 and H3K27me3 in the PROX1 promoter. (E) KSHV-infected PDLSCs co-transduced with si-vIL-6 and si-vGPCR were assayed for PROX1 expression by Western blot, and (F) the histone modifications in the PROX1 promoter by ChIP assay. (G and H) vGPCR or vIL-6/vGPCR expressing PDLSCs were incubated with a neutralizing antibody against VEGF-A (α-VEGF-A). The effects of the neutralizing antibody on cell angiogenesis and malignant invasion were examined by tubule formation assay (G) and Transwell-matrigel invasion assay (H), respectively. Statistics analysis was performed using the one-way ANOVA test, and P-value was calculated by GraphPad Prism. P-values < 0.05 were considered significant (*P<0.05, **P<0.01 and ***P<0.001) and N.S represented no significance.

(Fig 2). Therefore, the regulation of PROX1 expression by KSHV is a key event in transforming a KSHV-infected MSC to Kaposi's sarcoma. In addition, recent studies reveled another role of KSHV-promoted PROX1 activation in KS tumorigenesis that PROX1 enhances KSHV lytic replication and sustains the population of KSHV infected cells that would otherwise be lost as KS cells divide [41,42]. Overall, the revelation of epigenetic regulation of PROX1 in MSCs by KSHV in this study provided a novel insight into the MEndT process and KS tumorigenesis.

Herein, the PROX1 gene was found to reside in bivalent chromatin where activating histone H3 Lys4 trimethylation (H3K4me3) mark and repressive histone H3 Lys27 trimethylation (H3K27me3) mark co-exist on the same nucleosome. In 2006, a subset of promoters associated with both activating H3K4me3 and repressive H3K27me3 marks, referred to as "bivalent" chromatin domain, was discovered in mouse embryonic stem cells (ESCs) [16]. Such a signature bivalent chromatin appears to be more specific to the promoter of developmentally important transcription factors. The synchronous existence of H3K4me3 and H3K27me3 posits these transcription factor genes in a poised state, enabling them to be rapidly activated upon suitable developmental cues and environmental stimuli [17]. Our finding that the

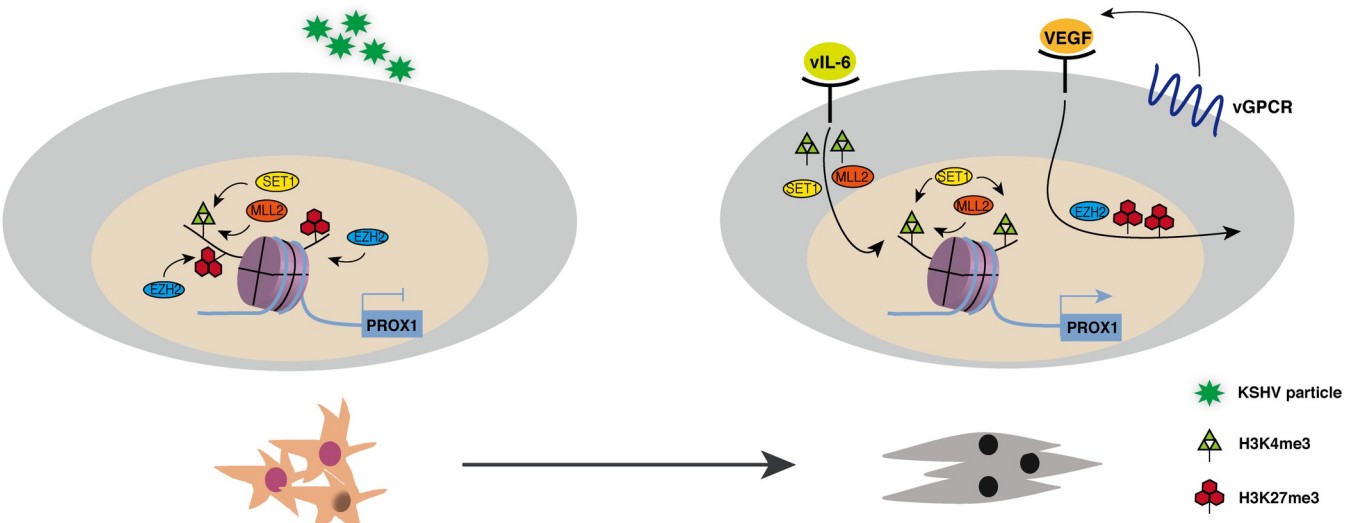

**Fig 8. Model for KSHV-mediated epigenetic regulation of MEndT through the bivalent chromatin of PROX1 gene.** PROX1 gene in MSCs harbors a distinctive "bivalent" epigenetic signature consisting of both active marker H3K4me3 and repressive marker H3K27me3. KSHV infection resolves the bivalent chromatin by decreased H3K27me3 and increased H3K4me3 to activate the PROX1 gene. vIL-6 signaling leads to the recruitment of MLL2 and SET1 complexes to the PROX1 promoter to increase H3K4me3, and the vGPCR-VEGF-A axis is responsible for removing PRC2 from the promoter to reduce H3K27me3. Therefore, through a dual signaling process, KSHV activates PROX1 gene and initiates MEndT, thereby transforming MSCs into KS-like spindle-shaped cells.

transcription factor PROX1 gene is also posited in the distinctive bivalent chromatin in oral MSCs suggests that lineage differentiation transcription factor genes in postnatal stem cells are regulated at the epigenetic level in the same mechanism as that used in embryonic stem cells. In this study, we investigated how KSHV resolved the bivalent chromatin domain for PROX1 activation and revealed that KSHV uses a dual signaling process, i.e., activating vIL-6 signaling to increase active histone marker H3K4me3 and using vGPCR-VEGF-A axis to reduce repressive histone marker H3K27me3, to resolve the bivalent chromatin structure and activate PROX1 gene expression. This is an interesting finding and suggests that there is a "two-factor-authentication" mechanism in the resolution of bivalent chromatin. The biological significance for such a two-factor-authentication has not yet been understood, but we speculate that genes in bivalent domain chromatin are poised for a quick response to an external stimulus. Still the dual signaling process may grant a more efficient and accurate (with an additional layer of security) response to activate the gene.

vIL-6 and vGPCR were found to play essential roles in resolving the bivalent chromatin domain and activating the PROX1 gene. vGPCR exerts this function through producing VEGF-A. Thus, vIL-6 and VEGF-A regulate PROX1 expression through an autocrine or paracrine mechanism. This finding explained our previous observations that MSCs can acquire KS tumorigenic properties by being incubated with conditioned media of KSHV-infected MSCs [32], and MSCs could undergo MEndT via paracrine regulation [6]. This finding also implies that KSHV may not only regulate PROX1 expression in its infected MSCs, but also modulate uninfected cells in the environment for PROX1 activation. In fact, it is well known that Kaposi sarcoma secrets abundant inflammation cytokines and growth factors including IL-1, IL-6, basic fibroblast growth factor (bFGF), platelet-derived growth factor (PDGF), tumor necrosis factor (TNF), interferon-gamma (IFN-γ), and vascular endothelial growth factor (VEGF), as well as KSHV-encoded vIL-6 [38,43–45]. In the early stage, the KS lesion is not a real sarcoma, but an angiohyperplastic-inflammatory lesion mediated by inflammatory cytokines and angiogenic factors. With the development of KS, spindle-shaped tumor cells accumulate and become dominant, triggered or amplified by KSHV infection [46]. Our result is consistent with the notion that KS is a cytokine disease and further suggests that cytokine autocrine or paracrine plays a central role in establishing KS spindle-shaped tumor cells in KS lesions.

## Materials and methods

### Ethics statements

The use of human samples and PDLSCs in this study was approved by the Medical Ethics Review Board of Sun Yat-sen University (approval no. 2015–028). Written informed consent was provided by study participants. The animal experiments were approved by the Animal Ethics Review Board of Sun Yat-sen University (approval no. SYSU-IACUC-2018-000162) and carried out strictly following the Guidance suggestion of caring laboratory animals, published by the Ministry of Science and Technology of the People's Republic of China.

### Cell culture

The isolation of human PDLSCs has been described previously [6]. PDLSCs were maintained in alpha minimal essential medium (α-MEM) with 10% heat-inactivated fetal bovine serum (FBS), 2 mM L-glutamine and antibiotics. The second to tenth passages of PDLSCs were used in this study. Human embryonic kidney HEK293T cells were purchased from American Type Culture Collection (ATCC) and cultured in Dulbecco's modified Eagle's medium (DMEM) supplemented with 10% FBS and antibiotics. iSLK.219 cells, cultured in DMEM with 10% FBS and antibiotics, were gifted from Dr. Ke Lan at Wuhan University.

## Reagents and antibodies

Cell culture medium (α-MEM and DMEM) and FBS were purchased from GIBCO Life Technologies. Penicillin and streptomycin were obtained from HyClone. Glutamine was obtained from Sigma. Matrigel for tube assay and cell invasion assay was obtained from Corning. Transwell inserts (PIXP01250) were purchased from Merck Millipore. Anti-PROX1 (BA2390) for Western blot and anti-PROX1 (ab199359) for immunohistochemistry and immunofluorescence were purchased from Boster and Abcam, respectively. Anti-LANA (ab4103) antibody for immunohistochemistry and immunofluorescence was purchased from Abcam. Antibodies against H3K4me3 (07–473) and H3K27me3 (07–449) were from Merck Millipore, and methyltransferases CFP1 (CXXC1,40672), EZH2 (5246), MLL2 (63735), WDR5 (13105) and RBBP5 (13171) were from Cell Signaling Technology, Inc. Antibody against UTX (ab36938) was from Abcam. Antibody against MLL1 (61295) was purchased from Active Motif. Human IL-6 (hIL-6) (14-8069-62) was from eBioscience, and VEGF-165 (100–20) was obtained from PeproTech, Inc. Antibody against VEGF-A (YT5108) was from ImmunoWay. PDPN (11629-1-AP), CD31 (11265-1-AP), VEGFR2 (26415-1-AP), VCAM1 (11444-1-AP) were from ProteinTech. LYVE-1 (sc-19316) and VEGFR3 (a5605) were obtained from Santa Cruz and Abclonal, respectively. Neutralizing antibody VEGF-A (11066-R010) were purchased from Sino Biological.

## KSHV preparation and infection

iSLK.219 cells carrying rKSHV.219 were induced for lytic replication by 1 μg/mL doxycycline and 1 mM sodium butyrate. Five days post-induction. The culture media were filtered through a 0.45 um filter and ultra-centrifuged with Beckman Coulter OptimaTM L-100XP at 100000 g for 1 h. The pellet was resuspended in 1/100 volume (culture media) of 1x PBS and stored at -80˚C until use. PDLSCs were seeded at $2x10^5$ cells per well in 6-well plates. Cells were infected with KSHV in the presence of polybrene (4 μg/mL) at an MOI of 50 (viral genome copy equivalent). After centrifugation at 2500 rpm for 60 min at room temperature, the cells were incubated at 37˚C with 5% $CO_2$ for 2 hours. Then, the inoculum was removed by changing culture medium.

## Chromatin immunoprecipitation (ChIP)

PDLSCs ($1x10^7$ cells) were cross-linked with formaldehyde of 1% final concentration for 15 min at room temperature, and the reaction was stopped by adding 125 mM glycine. Cells were washed with cold 1x PBS three times and collected by scratching the cells with cold 1x PBS. The cells were spun down under 2000 g at 4˚C, resuspended with 1 mL lysis buffer (1% SDS, 10 mM EDTA, 50 mM Tris-HCl PH7.0, freshly added 1 mM PMSF, complete protease inhibitor [Roche]) for 10 min at 4˚C and subjected to sonication to fragment the genomic DNA in size range of 300–700 base pair. Samples were centrifuged at 14000 g for 15 min at 4˚C, and supernatants were collected as soluble chromatin. For each ChIP assay, 100 μl of chromatin solution was diluted to 1 ml with dilution buffer (0.01% SDS, 1.1% Triton X-100, 1.2 mM EDTA, 20 mM Tris-HCl pH8.0, 167 mM NaCl, complete protease inhibitor [Roche]). Antibody (2 μg) was added to the chromatin sample and incubated overnight at 4˚C. Streptavidin magnetic beads (Merck Millipore), which had been washed three times in lysis buffer and blocked with 100 mg/mL sheared salmon sperm DNA (Ambion), were added to chromatin-antibody mixture, and incubated at 4˚C for 4 hours. The beads were washed 3 times with low salt (0.01% SDS, 1% Triton X-100, 2 mM EDTA, 20 mM Tris-HCl PH8.0, 150 mM NaCl), high salt (0.01% SDS, 1.1% Triton X-100, 1.2 mM EDTA, 20 mM Tris-HCl PH8.0, 500 mM NaCl), and LiCl (250 mM LiCl, 1% NP-40, 1% sodium deoxycholate, 1 mM EDTA, 20 mM Tris-HCl pH8.0) washing buffers, respectively, and two times with 1x TE buffer (100 mM Tris-HCl, pH8.0 and 10 mM EDTA, pH8.0). DNA was eluted from the beads with 400 μl

elution buffer (1% SDS, 0.1 M NaHCO$_3$). Immune complexes were eluted by 150 ml TE buffer (pH8.0) and cross-links were reversed by heating at 65˚C overnight, followed by digestion with Proteinase K (0.2 U/mL) at 50˚C for 2 h. DNA was purified with Magen HiPure DNA Clean Up Kit (D2141) and analyzed by real-time PCR with primers for the PROX1 promoter.

## Sequential Chromatin immunoprecipitation (sequential ChIP)

Sequential ChIP was performed as previously described [16]. Briefly, the first step was standard ChIP with antibody against H3K27me3 and the sheared DNA fragments were controlled in the range of 400–500 base pairs. Eluted chromatin was diluted 50-fold and subjected to a second immunoprecipitation with antibody against H3K4me3. Immune complexes were eluted, cross-linking was reversed, and DNA was purified as described above.

## Quantitative PCR (qPCR) and reverse transcription- qPCR (RT-qPCR)

PCR primers for evaluating ChIP or sequential ChIP assays were designed to amplify 150–200 base pair fragments from the promoter regions as follows: PROX1 (forward: 5'-GACCCCCAG ATTCCCAGGTCCTTCT-3'; reverse: 5'-AAGCCAGATTTCTATATTTTTTCTG-3'), PML (forward: 5'-TTTCGGACAGCTCAAGGGAC-3'; reverse: 5'-TTAGTTTCGATTCTCGGTTT-3'), TGFBR2 (forward: 5'-AGCTGTTGGCGAGGAGTTTC-3'; reverse: 5'-AGGAGTCCGGC TCCTGTCCC-3'), and TGFB3 (5'-GGGAGTCAGAGCCCAGCAAA; reverse: 5'-TGGCA ACCCTGAGGACGAAG-3'). PROX1 coding sequence region primer (forward: 5'-GAG CCCTGATCAGAGAGCAGGAAA; reverse: 5'-GACTTTGACCACAGTGTCCACAAC). Real-time PCR was carried out using SYBR Green PCR mix (Roche) in Roche LightCycler 480II Instrument.

RNA was isolated using an Ultrapure RNA Kit (CWBIO CW0581), reverse transcribed (Takara), and quantified using SYBR green PCR master mix on a Roche LightCycler 480II. The following primers (5'-3') were used: VEGF-A (forward: AGGGCAGAATCATCACGA AGT; reverse: AGGGTCTCGATTGGATGGCA), VEGF-B (forward: GAGATGTCCCTGGA AGAACACA; reverse: GAGTGGGATGGGTGATGTCAG), VEGF-C (forward: GAGGAGC AGTTACGGTCTGTG; reverse: TCCTTTCCTTAGCTGACACTTGT), VEGF-D (forward: ATGGACCAGTGAAGCGATCAT; reverse: GTTCCTCCAAACTAGAAGCAGC), vIL-6 (forward: TCGTTGATGGCTGGTAG; reverse: CACTGCTGGTATCTGGAA), and vGPCR (forward: AACCATCTTCTTAGATGATGAT; reverse: AATCCATTTCCAAGAACATTTA).

## Immunohistochemistry (IHC) analysis

Paraffin-embedded Kaposi sarcoma samples and kidney capsule transplants were sectioned. Deparaffinated sections were subjected to hematoxylin and eosin (H&E) staining (Servicebio G1002). For IHC staining, sections were subjected to antigen retrieval using 10 mM sodium citrate buffer (pH 6.0) for 10 min with an electric pressure cooker. Then tissues were treated in 3% hydrogen peroxide for 10 min to quench the endogenous peroxidase activity and probed with antibodies against anti-PROX1 (1:200), LANA (1:150), CD31 (1:100), PDPN (1:100), or VCAM1 (1:100) at 4˚C overnight. The primary antibody binding was detected using a goat anti-rabbit HRP secondary antibody (Maxim DAB-1031), followed by colorimetric detection using metal enhanced DAB. Tissues were counterstained with hematoxylin.

## Immunofluorescence assay

Cells were fixed with 3.6% formaldehyde in PBS for 10 min, permeabilized in 0.1% Triton X-100 for 15 min and blocked in 1% BSA for 1 h at room temperature. The samples were

incubated with antibodies against LANA (1:200) or PROX1(1:200) respectively at 4˚C over-night. Goat anti-Rat IgG Alexa Fluor 555 (1:200; Invitrogen A-21434) and Goat anti-Rabbit IgG Alexa Fluor 647 (1:200; Invitrogen A-21244) were used as secondary antibodies. Nuclei were stained by Hoechst 33258 (Sigma) for 3 min at room temperature. Slides were examined under a Nikon pE-300 fluorescent microscope (400x), and three channels were recorded sequentially.

## Western blot

Cells were lysed on ice for 30 min with lysis buffer (50 mM Tris-HCl pH 7.4, 150 mM NaCl, 1% NP-40, 1 mM NaF, 1 mM $Na_3VO_4$, 1 mM PMSF, protease inhibitor cocktail [1 tablet in 50 mL lysis buffer, Roche]). The whole cell extract was denatured, resolved by SDS-PAGE, and transferred onto nitrocellulose membranes. The membranes were blocked in 5% nonfat milk in 1xTris-buffered saline (TBS) for 1 h and incubated with diluted primary antibodies PROX1 (1:1000), PDPN (1:1000), VEGFR2 (1:500), VEGFR3 (1:500), LYVE-1 (1:100), or VEGF-A (1:1000) at 4˚C overnight. β-Actin (1:5000; Sigma A5441) served as the internal reference. IRDye 680LT/800CW goat anti-rabbit or anti-mouse antibody (Li-Cor Biosciences) was used as a secondary antibody. The blots were visualized in an Odyssey system (Li-COR).

## shRNA-mediated knockdown of PROX1 expression

An shRNA lentiviral vector targeting the CDS region of PROX1 mRNA (Clone ID: NM_002763.3-531s21c1) was purchased from Sigma-Aldrich. Lentiviral particles were pre-pared by transfecting HEK293T cells with pLKO.1-shPROX1 (or pLKO.1-shControl), psPAX2, pMD2.G plasmids at the ratio of 5:3:2. Media containing lentiviruses were harvested at 48 h and 72 h and used to transduce PDLSCs. Transduced PDLSCs were selected under 2 μg/mL puromycin for a week.

## Tube formation assay

48-well plates were pre-coated with matrigel (1:1 dilute with α-MEM without FBS, 100 μL/ well) and incubated at 37˚C for 1 h to allow gelation to occur. $1x10^5$ PDLSCs were suspended in 200 μL α-MEM without FBS and placed to the top of the gel. The cells were incubated at 37˚C with 5% $CO_2$ for 6–8 h, and tube formation images were captured using a ZEISS micro-scope. The quantitation of the tube was carried out using the ImageJ software to measure the length of the total tube segments in the image. The average value was used for the histogram.

## Cell invasion assay

Cell invasion assays were carried out in 24-well Transwell units. Briefly, polycarbonate filters with 12-mm pores were precoated with 100 μL of matrigel (1:10 diluted with α-MEM without FBS). $1.5x10^4$ cells in serum-free media were placed in the top wells, and the bottom chambers were filled with 20% FBS medium. After 24 h incubation, the cells that had passed through the filter were stained with crystal violet. The number of invaded cells was counted from multiple randomly selected fields using ImageJ software. Photographs were taken and independent experiments were performed in triplicate.

## Ectopic expression of viral genes

KSHV open reading frames including K1, K12, ORF45, ORF50/RTA, ORF71/vFLIP, ORF73/ LANA, K8, PAN, vIL-6, and vGPCR were cloned into the pMSCVpuro vector at the Bgl II and EcoRI sites using ClonExpression II One Step Cloning Kit (Vazyme C112-02). Retroviral

particles were prepared by transfecting HEK293T cells with pMSCVpuro-(K1, K12, ORF45, ORF50/RTA, ORF71/vFLIP, ORF73/LANA, K8, PAN, vIL-6, and vGPCR) and PIK plasmids at the ratio of 1:1. Media containing retroviruses were harvested at 48 and 72 h and used to transduce PDLSCs. Transduced PDLSCs were selected under 2 μg/mL puromycin for a week.

## siRNA-mediated silence of vIL-6 and vGPCR

Small interference RNAs targeting KSHV encoded interleukin-6 (vIL-6) and G-protein coupled receptor(vGPCR) were obtained from Guangzhou RiboBio Co., Ltd. Si-vIL6 (5'-TGTTCTGAGTGCAATGGAA-3') and si-vGPCR (5'-CCTCATAAATGTTCTTGGA-3') (or si-NC, si-GAPDH, Cy3) were transiently transfected into KSHV-PDLSCs using Lipofectamine 3000 Transfection Regent (Invitrogen L3000015). The transfection efficiency of siRNA was assessed according to Cy3 under a fluorescence microscope, and transfected cells were ready for the next experiment 48–72 h post-transfection.

## Supernatant transfer assay

The supernatant from KSHV-PDLSCs (48 hpi) was transferred to PDLSC culture at the ratio of 1:1. HEK293T cells were transfected with vIL-6 expression vector, and media were collected after 48 h post-transfection 48. The supernatant with vIL-6 was added to PDLSC culture at the ratio of 1:1.

## Kidney capsule transplantation

PDLSCs were suspended with α-MEM media and seeded on 96-well plates precoated with 0.5% agarose at $2x10^4$ cells per well. The spheroids were grown at 37˚C for 1 to 2 d with 5% humid $CO_2$. The media containing spheroids were collected and transferred to 15 mL conical tubes, washed twice with 1x PBS, and centrifuged at 1000 rpm for 5 min. About 100 spheroids were placed on gelfoam scaffold for 1 d in medium. Recipient female nude mice (6–8 weeks, n = 3–5) were weighed and anesthetized with isoflurane. Kidney was exposed through an incision of the skin and muscle on the back of the mouse. The kidney capsule was opened with the fine tip of no.5 forceps. The spheroids/scaffold block was placed under the kidney capsule. Sutures were placed, and capsule-grafting products were harvested 28 d after transplantation.

## Statistical analysis

Data were analyzed by two-tailed Student's t-test and one-way ANOVA in GraphPad Prism. P values < 0.05 were considered significant (*$P < 0.05$, **$P < 0.01$ and ***$P < 0.001$) and N.S represented no significance.

## Supporting information

**S1 Fig. Histone modifications (H3K4me3 and H3K27me3) in the PROX1 coding region in mock- and KSHV-infected PDLSCs.** Mock- and KSHV-infected PDLSCs were subjected to a Chromatin immunoprecipitation (ChIP) assay with antibodies against H3K4me3 and H3K27me3. The extracted DNA was analyzed by qPCR with primers of the PROX1 coding region.
(TIF)

**S2 Fig. Expression of vIL-6, vGPCR, and vFLIP in PDLSCs transfected with their expression vectors.** PDLSCs were transfected with a class of viral gene expression vectors, including K1, K12, ORF45, ORF50, ORF71, ORF73, K8, PAN, vIL-6, and vGPCR (Fig 5). Among them, vIL-6, vGPCR and vFLIP were tagged with HA or Flag tags. The expression of vIL-6, vGPCR

and vFLIP were examined by Western blot.
(TIF)

**S3 Fig. Blockade of vGPCR-VEGF-A axis using a neutralizing antibody against VEGF-A inhibits PROX1 activation and KSHV-promoted angiogenesis and invasion phenotypes.** (A) PDLSCs were treated with conditioned medium from KSHV-PDLSC. The expression of PROX1 and PDPN were analyzed by Western blot. (B) KSHV-infected PDLSCs (K-PDLSCs) were placed on Matrigel, and tubulogenesis was examined by Tubule formation assay and quantified by measuring the total segment tube length using the ImageJ software. (C) K-PDLSCs ($1.5 \times 10^4$ cells/well) were seeded in the upper chamber of Transwell with Matrigel in serum-free α-MEM. α-MEM containing 20% FBS was used as a stimulus for chemotaxis in the lower chamber. Cells migrated to the lower chamber were stained with crystal violet. The error bars represent SD. Statistics analysis was performed using t-test in GraphPad Prism. P-values < 0.05 were considered significant (*P<0.05, **P<0.01 and ***P<0.001) and N.S represented no significance.
(TIF)

## Acknowledgments

We thank all members of Yuan Lab for critical reading of this manuscript and constructive suggestions.

## Author Contributions

**Conceptualization:** Yao Ding, Yan Yuan.

**Data curation:** Yao Ding.

**Formal analysis:** Yao Ding.

**Funding acquisition:** Yan Yuan.

**Investigation:** Yao Ding, Weikang Chen, Zhengzhou Lu.

**Methodology:** Yao Ding, Weikang Chen, Zhengzhou Lu, Yan Wang.

**Resources:** Yan Wang.

**Supervision:** Yan Wang.

**Writing – original draft:** Yao Ding.

**Writing – review & editing:** Yan Yuan.

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
