## [Decision Letter · Decision Letter 0]

7 Jun 2021

Dear Yan,

Thank you very much for submitting your manuscript "Kaposi's Sarcoma-associated Herpesvirus Promotes Mesenchymal-to-Endothelial Transition by Resolving the Bivalent Chromatin of PROX1 Gene" for consideration at PLOS Pathogens. As with all papers reviewed by the journal, your manuscript was reviewed by members of the editorial board and by several independent reviewers. The reviewers appreciated the attention to an important topic. Based on the reviews, we are likely to accept this manuscript for publication, providing that you modify the manuscript according to the review recommendations.

Sincerely,

Pinghui Feng

Associate Editor

PLOS Pathogens

Blossom Damania

Section Editor

PLOS Pathogens

Kasturi Haldar

Editor-in-Chief

PLOS Pathogens

orcid.org/0000-0001-5065-158X

Michael Malim

Editor-in-Chief

PLOS Pathogens

orcid.org/0000-0002-7699-2064

Reviewer Comments (if any, and for reference):

Reviewer's Responses to Questions

**Part I - Summary**

Reviewer #1: This study described very interesting observation, which KSHV infection induces the mesenchymal-to-endothelial transition (MEndT) through induction of PROX1. PROX1 has recently been described as an important cellular factor for maintaining lytic phenotype of KSHV infection (Gramolelli S. et al., Cancer Res 2020; Choi D. et al., 202 Cancer Res 2020), and this study further adds to the previous studies that KSHV encoded proteins could induce PROX1 genes via increasing H3K4me3 and decreasing H3K27me3 marks on PROX1 promoter; this mechanism may establish suitable environment for maintaining KSHV life cycle. The authors also proposed interesting epigenetic model of “two-factor-authentication” for efficient and accurate response for activation of the putative pioneer factor. VIL-6 and vGPCR were found to be responsible for PROX1 up-regulation through paracrine fashion. Although number of samples was limited, the authors also included patients samples for the IHC staining to further demonstrate upregulation of PROX1 at KS lesions. This reviewer think that this study is conceptually novel, experiments are well-designed and controlled, conclusion is supported by the results.

Reviewer #2: In this manuscript, Yao et al. revealed that KSHV infection of MSCs induced the expression of PROX1, which plays a critical role in the process of Mesenchymal-to-Endothelial transition. Mechanistically, the authors found that KSHV infection promoted the expression of PROX1 by resolving the bivalent chromatin; vIL-6 increased the active histone marker H3K4me3 by recruiting MLL2 and Set1 complex, and vGPCR-VEGF-A reduced the repressive marker H3K27me3 by removing EZH2. Overall, the study provides a novel model for the epigenetic regulation of Prox1 by KSHV and most data are convincing and well presented.

Reviewer #3: Understanding the molecular mechanisms of Kaposi’s sarcomagenesis in the right cellular context is still an important gap in our knowledge of KSHV oncobiology. The present manuscript represents a significant and novel contribution to this field. The authors have previously shown that infection of KSHV of human mesenchymal stem cells leads to cellular reprogramming consistent with a Mesenchymal to Endothelial transition (MEndT). These findings and those of other labs have substantiated the possibility that the MSC is a bona fide KSHV target and KS spindle cell progenitor. In the current manuscript, the authors seek to understand the upregulation of PROX1, a master regulator of lymphatic endothelial differentiation and reprogramming by KSHV infection. They show that in consistence with their results on the ability of KSHV to reprogram MSC towards a lymphatic endothelial lineage, KSHV infection upregulates PROX1. Using silencing, they show that PROX1 is essential for the expression of endothelial and lymphatic endothelial lineage markers. Then they carry out a sequential CHIP analysis to determine the epigenetic regulation of PROX1 by KSHV. They found it involves removing repressive histone modifications (H3K27me3) and activating histone modification (H3K4me3) marks. Finally, they screen potential KSHV genes that could regulate PROX1 epigenetically. They identify vIL-6 and vGPCR as genes able to upregulate PROX1. They define that vIL-6 can act directly, while vGPCR acts via a VEGF-VEGRFR autocrine loop, showing they can also act paracrinally. Using CHIP, they identify vIL-6 responsible for MLL and Set1 mediated H3K4me3 activation and vGPCR as responsible for removing PRC repressive marks (H3K27me3). Using silencing, they show that depletion of these genes decreases the PROX-1 promoted endothelial activities. This paper is a significant contribution to our understanding of the molecular mechanisms of KSHV oncogenesis in the context of mesenchymal stem cell progenitors. It reinforces the critical and necessary role of KSHV angiogenic oncogenes such as vIL-6 and vGPCR in the early stages of KSHV sarcomagenesis.

Enrique A. Mesri

**Part II – Major Issues: Key Experiments Required for Acceptance**

Reviewer #1: 1. Depending on resolution of ChIP-assay (fragmentations by sonication), the authors theory for “two-factor-authentication” may not be accurate. Prox1 promoter and a lncRNA, Prox1-AS, promoter localize very next each other (maybe partially overlapped), and very strong H3K27me3 peaks are often adjacent to strong H3K4me3 peaks in public ChIP-seq data sets (including both PEL and iSLK cells). There is a possibility that one promoter is regulated by vIL-6 and the other is regulated by GPCR pathway (not same nucleosome), and ChIP-assays with fixed DNA/histone complex might be precipitating adjacent nucleosomes. Of course, there is a possibility that PDLSCs indeed possess the bivalent histone mark and regulates the PROX1 promoter as the author suggested. For this reviewer, it is a little difficult to understand that activation of a nucleosome, which usually increases H3K27 acetylation via coactivator complex recruitment, did not reduces H3K27me3. Examining Prox1-AS expression in presence of vIL-6 and/or vGPCR, or designing additional primer sets that probe Prox1-AS promoter would clarify the result, if the authors like to do so. In minimum, the authors may expand the discussion to avoid a possible misleading.

2. For ChIP-assays, it would be important to include other genomic regions as controls in order to demonstrate relative enrichment of H3K4me3 and H3K27me3 marks on the PROX1 promoter.

3. Fig. 3C, do the authors expect to see stronger H3K4me3 signals in KSHV infected cells (K27/K4)? Please clarify. Line 721 for the legend, it should read qPCR (not RT-qPCR).

Reviewer #2: This reviewer has a few concerns and suggestions listed below.

1. All WB crops are lack of molecular weight labeling.

2. Some statistical data are lack of significance evaluation (e.g., Fig 2C and D, list is not exhaustive.)

3. Fig5A, it would be more convincing if authors could show the expression data of the indicated KSHV genes. ORF71 seemed to slightly induce PROX1 and greatly promoted the expression of PDPN, while vIL-6 and vGPCR affected Prox1 more specifically. Authors may provide some explanation.

4. Figure 7G and H showed the effect of ectopically expressed vIL-6 and vGPCR, and it would be better to present these data in Figure 6. On the other hand, here authors would better show how ablation of vIL-6 and/or vGPCR in K-PDLSC affects tube formation and invasion.

5. Fig7C, vGPCR is a lytic gene and this reviewer noted that the knockdown was relatively inefficient. Authors may consider including PDLSC control to show that vGPCR is indeed expressed in K-PDLSC.

6. Although authors showed that siRNA-mediated knockdown of vIL-6 and vGPCR abolished the expression of Prox1 in K-PDLSC, it would be more convincing if authors could use knockout virus to confirm the observation.

Reviewer #3: The paper will benefit from some figure improvements that will strengthen its presentation.

In Figure 6 it will be interesting to show as much as possible the experimental systems of Figure 1 to show the extent of the contributions of vIL-6 and vGPCR to the endothelial-lineage and LEC phenotypes shown in Figure 1, particularly the western blots showing the endothelial markers.

The regulation of PROX1 by vGPCR via VEGF is interesting, particularly in published evidence showing that vGPCR signaling and oncogenicity become potentiated in the LEC lineage. It will be interesting to determine whether there is a mutual regulation of vGPCR by PROX1 in this system as well.

The autocrine mechanisms proposed in Figure 6 are intriguing. These data would be reinforced by using blocking antibodies to the VEGF and IL-6 receptors.

**Part III – Minor Issues: Editorial and Data Presentation Modifications**

Reviewer #1: 1. Please double check accuracy of references. For example, Reference 20 (Page 6) does not match what the paper describes.

2. It is important to include two recent studies regarding function of PROX1 on KSHV lytic replication (above mentioned). These studies have direct association with the authors studies and add an exciting discussion point, which KSHV establishes suitable microenvironment for KSHV replication through PROX1 activation via vIL-6 and vGPCR.

Reviewer #2: (No Response)

Reviewer #3: The quality of some western blots that reflects the impact of silencing in key phenotypes may be improved, and its quantitative information could be reinforced by densitometric quantification of duplicates or triplicates.

PLOS authors have the option to publish the peer review history of their article (what does this mean?). If published, this will include your full peer review and any attached files.

Reviewer #1: **Yes: **Yoshihiro Izumiya

Reviewer #2: No

Reviewer #3: **Yes: **Enrique A. Mesri

Figure Files:

Data Requirements:

Reproducibility:

References:

---

## [Decision Letter · Decision Letter 1]

27 Jul 2021

Dear Yan,

We are pleased to inform you that your manuscript 'Kaposi's Sarcoma-associated Herpesvirus Promotes Mesenchymal-to-Endothelial Transition by Resolving the Bivalent Chromatin of PROX1 Gene' has been provisionally accepted for publication in PLOS Pathogens.

Best regards,

Pinghui Feng

Associate Editor

PLOS Pathogens

Blossom Damania

Section Editor

PLOS Pathogens

Kasturi Haldar

Editor-in-Chief

PLOS Pathogens

orcid.org/0000-0001-5065-158X

Michael Malim

Editor-in-Chief

PLOS Pathogens

orcid.org/0000-0002-7699-2064

Reviewer Comments (if any, and for reference):

Reviewer's Responses to Questions

**Part I - Summary**

Reviewer #1: The authors addressed this reviewer's concerns.

Reviewer #2: The authors have addressed all my concerns and the manuscript is significantly improved. I have no further comments.

Reviewer #3: The authors have successfully addressed most of all the three reviewer's comments with comments and or new data. The paper in its revised form is an important contribution to the understanding of epigenetic mechanisms occurring during KSHV mesenchymal sarcomagenesis and its relationship with the endothelial lineage.

Enrique Mesri

**Part II – Major Issues: Key Experiments Required for Acceptance**

Reviewer #1: (No Response)

Reviewer #2: (No Response)

Reviewer #3: (No Response)

**Part III – Minor Issues: Editorial and Data Presentation Modifications**

Reviewer #1: Followings are a few minor points that the authors may want to double check.

1. Some actin blots look a little strange. It seems upside down. (for example Fig. 6D, 7C)

2. ChiP-assays in Fig. 3A. Enrichment of H3K4me3 marks on TGFB3 promoter is less than PROX1 IgG control. The authors are comparing with different genomic regions by using same antibody with input as normalization. If this is the case, do the authors consider that the TGFB3 promoter region is occupied with H3K4me3 modified nucleosomes? This review think that including absolute positive and negative genomic regions (e.g. Actin promoter) may help to clarify.

Reviewer #2: (No Response)

Reviewer #3: (No Response)

PLOS authors have the option to publish the peer review history of their article (what does this mean?). If published, this will include your full peer review and any attached files.

Reviewer #1: **Yes: **Yoshihiro Izumiya

Reviewer #2: **Yes: **Junjie Zhang

Reviewer #3: **Yes: **Enrique A. Mesri

---

## [Editor Report · Acceptance letter]

23 Aug 2021

Dear Dr. Yuan,

We are delighted to inform you that your manuscript, "Kaposi's Sarcoma-associated Herpesvirus Promotes Mesenchymal-to-Endothelial Transition by Resolving the Bivalent Chromatin of PROX1 Gene," has been formally accepted for publication in PLOS Pathogens.

Best regards,

Kasturi Haldar

Editor-in-Chief

PLOS Pathogens

orcid.org/0000-0001-5065-158X

Michael Malim

Editor-in-Chief

PLOS Pathogens

orcid.org/0000-0002-7699-2064